# Heat Shock Transcription Factor GhHSFB2a Is Crucial for Cotton Resistance to *Verticillium dahliae*

**DOI:** 10.3390/ijms24031845

**Published:** 2023-01-17

**Authors:** Lu Liu, Qi Wang, Linfeng Zhu, Huiming Guo, Hongmei Cheng, Xiaofeng Su

**Affiliations:** 1Biotechnology Research Institute, Chinese Academy of Agricultural Sciences, Beijing 100081, China; 2Agricultural Information Institute, Chinese Academy of Agricultural Sciences, Beijing 100081, China; 3Hainan Yazhou Bay Seed Lab, Sanya 572024, China

**Keywords:** cotton, *Verticillium dahliae*, *GhHSFB2a*, RNA-seq, resistance

## Abstract

Heat shock transcription factors (HSFs) play a critical regulatory role in many plant disease resistance pathways. However, the molecular mechanisms of cotton HSFs involved in resistance to the soil-borne fungus *Verticillium dahliae* are limited. In our previous study, we identified numerous differentially expressed genes (DEGs) in the transcriptome and metabolome of *V. dahliae*-inoculated *Arabidopsis thaliana*. In this study, we identified and functionally characterized *GhHSFB2a*, which is a DEG belonging to HSFs and related to cotton immunity to *V. dahliae*. Subsequently, the phylogenetic tree of the type two of the HSFB subfamily in different species was divided into two subgroups: *A. thaliana* and strawberry, which have the closest evolutionary relationship to cotton. We performed promoter cis-element analysis and showed that the defense-reaction-associated cis-acting element-FC-rich motif may be involved in the plant response to *V. dahliae* in cotton. The expression pattern analysis of *GhHSFB2a* displayed that it is transcriptional in roots, stems, and leaves and significantly higher at 12 h post-inoculation (hpi). Subcellular localization of GhHSFB2a was observed, and the results showed localization to the nucleus. Virus-induced gene silencing (VIGS) analysis exhibited that *GhHSFB2a* silencing increased the disease index and fungal biomass and attenuated resistance against *V. dahliae*. Transcriptome sequencing of wild-type and *GhHSFB2a*-silenced plants, followed by Gene Ontology, Kyoto Encyclopedia of Genes and Genomes, protein–protein interaction, and validation of marker genes revealed that ABA, ethylene, linoleic acid, and phenylpropanoid pathways are involved in GhHSFB2a-mediated plant disease resistance. Ectopic overexpression of the *GhHSFB2a* gene in *Arabidopsis* showed a significant increase in the disease resistance. Cumulatively, our results suggest that *GhHSFB2a* is required for the cotton immune response against *V. dahliae*-mediated ABA, ethylene, linoleic acid, and phenylpropanoid pathways, indicating its potential role in the molecular design breeding of plants.

## 1. Introduction

Cotton (*Gossypium hirsutum*) is the most important natural textile fiber worldwide, accounting for approximately 35% of the annual global fiber demand [1]. The losses in cotton yield and quality are mainly caused by Verticillium wilt (VW), a vascular disease induced by the soil-borne pathogen *Verticillium dahliae* [2]. *V. dahliae* can survive in the soil as microsclerotia for a long time, colonize through the roots of cotton, and invade the vascular bundles, causing the yellowing and wilting of cotton leaves, and even leading to plant death [3]. VW causes more than 30% of the annual cotton output to be lost in China, and to date, no effective fungicides have been developed [4]. Therefore, cultivating new disease-resistant varieties has become the most economical means of reducing damage caused by VW [5].

Transcription factors participate in plant developmental processes and responses to environmental change [6]. Among them, heat shock transcription factors (HSFs) mainly regulate plant adaptation to biotic and abiotic stresses [7,8]. HSFs bind to a heat shock element (HSE) in the promoter of downstream stress response genes to exert their role. On the basis of the structural characteristics of HSFs [9,10], proteins are classified into three major subfamilies: HSFA, HSFB, and HSFC [11]. HSFA and HSFC [12,13,14,15] are primarily involved in the regulation of high-temperature stress and salt tolerance [16,17,18].

Current studies on HSFB have focused on regulating abiotic stresses, such as high temperature, cold, and drought [19,20,21]. Overexpression of *TaHSF3* increased survival rate and improves cold tolerance in *Arabidopsis thaliana* [13]. In japonica rice, *OsHsfB2b* induced substantial upregulation under high-temperature conditions. Meanwhile, its overexpression resulted in decreased proline and malondialdehyde content, and relative conductivity [22].

In recent years, there has been some progress in understanding the role of HSFB in resistance to biotic stresses. Different *OsHSFB* members showed different expression patterns when rice was infested with *Xanthomonas campestris*. *HsfB1* and *HsfB2b* are involved in *pdf1.2* expression and disease-resistance response, and overexpression of *HsfB1* can lead to cell death [23]. *OsHsfB4b* regulated the disease defense response by binding to specific HSE [24]. Although several studies have shown that *HSFB* participates in pathogenic fungal stress, it is unclear whether the cotton *HSFB* gene is involved in the defense against *V. dahliae* and the underlying mechanisms.

In a previous study, we investigated the transcriptome and metabolome of *Arabidopsis* from pre-infected material in the laboratory using *V. dahliae* and screened for many differentially expressed genes (DEGs) through a series of bioinformatic analyses. Among them, we found that a *G. hirsutum HSFB*-2a-like gene, designated *GhHSFB2a* (LOC107904615), is highly homologous to a DEG. *GhHSFB2a* is transcriptionally induced by *V. dahliae* and is ubiquitous in cotton. The GhHSFB2a protein was found to be localized to the nucleus in a subcellular localization assay. Virus-induced gene silencing (VIGS) of *GhHSFB2a* significantly impairs cotton resistance against *V. dahliae*. The transcriptome of *GhHSFB2a*-silenced plants was comprehensively analyzed using an RNA-seq. The results showed that the downregulation of *GhHSFB2a* led to changes in the marker genes of ABA, ethylene, linoleic acid, and phenylpropanoid pathways. Likewise, the ectopic expression of *GhHSFB2a* in *Arabidopsis* obviously enhanced resistance against *V. dahliae* compared to the wild type (WT). The above results show that the *GhHSFB2a* gene played an important role in cotton resistance to *V. dahliae* by regulating plant hormones and secondary metabolites, which provided more abundant germplasm resources for the control of VW.

## 2. Results

### 2.1. Bioinformatics Analysis of GhHSFB2a

The gene designated *GhHSFB2a* was located on chromosome D11 of *G. hirsutum*, which contains two exons and one intron, encoding a 326 amino acid protein with an estimated molecular mass of 36.246 kDa and an isoelectric point of 5.93. Protein sequence analysis showed that GhHSFB2a contains a DNA-binding domain (DBD) between amino acids 111 and 168. Combined with multiple sequence alignment and 3D structure prediction results, the HSFB protein in *G. hirsutum* had a helix-turn-helix (H2-T-H3) motif in the DBD hydrophobic center (Figure 1a,b). The evolutionary relationships of the HSFB2 subfamily between cotton and common dicotyledonous plants, such as *A. thaliana*, tomato, apple, and strawberry, and important crops, such as rice and maize, were analyzed (Figure 1c).

The phylogenetic tree could be classified into two subgroups and revealed that *A. thaliana* and strawberry had a close evolutionary relationship with cotton, and the similarity between *GhHSFB2a* and *AtHSFB2a* or *FvHSFB2a* was 59.44 and 55.99%, respectively. The 2000 bp sequence upstream of the *GhHSFB2a* gene was predicted for *cis*-elements to analyze its regulatory relationship at the transcriptional level. The *GhHSFB2a* promoter region contains some basic elements, such as CAAT-box and TATA-box; endogenous hormone elements, such as acid corresponding elements ABRE4 and ABRE; gibberellin response elements GARE-motif; and immunity or defense-reaction-associated cis elements TC-rich-motifs. This suggests that *GhHSFB2a* may be regulated by environmental factors and endogenous hormones and may participate in the stress response (Figure 1d).

### 2.2. Transcriptional Levels of GhHSFB2a Were Upregulated under V. dahliae Stress

To explore the role of *GhHSFB2a* in cotton resistance to *V. dahliae*, we detected the expression pattern of *GhHSFB2a* in *V. dahliae*-infected cotton plants. Samples were collected from different tissues at 0–72 h post indentation (hpi). The results revealed that *GhHSFB2a* was expressed in all tissues, including the roots, stems, and leaves. *GhHSFB2a* expression was considerably upregulated at 0.5 hpi, after a slight decrease, and reached the highest level at 12 hpi, then gradually decreased (Figure 2). This reveals that *GhHSFB2a* is involved in cotton resistance to *V. dahliae*.

### 2.3. GhHSFB2a Is Localized in the Nucleus

The subcellular localization of GhHSFB2a was predicted to be in the nucleus. To further confirm the subcellular localization of the GhHSFB2a protein, full-length *GhHSFB2a* was fused to the pYBA1132*-eGFP* vector and transiently co-expressed in *Nicotiana benthamiana* leaves with a nuclear localization marker (*H2B-mCherry*) via agroinfiltration [25]. The localization of *GhHSFB2a* was observed in the lower epidermal cells of tobacco 48 h post-infiltration. As a control, fluorescence was observed uniformly and diffusely distributed in the cytoplasm, plasma membrane, and nucleus, which co-transformed with pYBA1132*-eGFP* and *H2B-mCherry*. In the upper panel, the GFP protein was localized in the whole cell, and the GFP-fused GhHSFB2a protein and H2B-mCherry were colocalized in the nucleus. These results suggested that the GhHSFB2a protein was localized in the nucleus and may function as a transcription factor that participates in the cotton immune response against *V. dahliae* infection (Figure 3).

### 2.4. Knockdown of GhHSFB2a Increases the Susceptibility of Cotton to V. dahliae

Loss-of-function experiments in cotton seedlings were performed to investigate the role of *GhHSFB2a* in immunity against *V. dahliae*, using the VIGS assay. We generated the recombinant pTRV2::*GhHSFB2a* construct and inoculated it into seedlings when the two cotyledons were fully developed. The emergence of photobleaching on the pTRV2::*CLA1* cotton plants was regarded as the success of the experiment (Figure 4A). A RT-qPCR was used to determine the expression levels of pTRV2::*GhHSFB2a* and pTRV2::*00* in plants. Transcript levels of *GhHSFB2a* were significantly reduced compared to those in control plants, validating the successful knockdown of the gene (Figure 4b). For the WT, the pTRV2::*00*, *GhHSFB2a*-silenced cotton was infected with V991, and water treatment was used as a control. Cotton disease progression was observed after V991 inoculation for two weeks, and the *GhHSFB2a*-silenced cotton exhibited evident stunting and wilting leaves, and the DI increased significantly. Cotton stems dissected longitudinally showed more pronounced browning of the vascular tissue compared to the WT cotton (Figure 4c,d). The fungal biomass was obviously increased in *GhHSFB2a*-silenced cotton compared to that in WT (Figure 4e). These results indicated that silencing *GhHSFB2a* increased fungal colonization and attenuated resistance against *V. dahliae*.

### 2.5. GhHSFB2a Regulates Resistance via Plant Hormones and Secondary Metabolites Pathways

To further elucidate the molecular function of *GhHSFB2a* in VW resistance, we performed the RNA-seq to explore the genome-wide changes between silenced and control plants. After filtering the reads with adapter, unknown reads, and low-quality reads, we obtained 39.54 Gb of clean data with an average GC content of 43.97. A total of 2549 DEGs with a maximum differential factor of 14, including 1354 upregulated and 1195 downregulated DEGs (Figure 5a), were observed. Cluster analysis was performed to determine the expression patterns of all DEGs (Figure 5b). We found that a variety of transcription factor genes closely related to plant disease resistance were differentially expressed, including *WRKY*, *MYB*, *bZIP*, and *AP2/ERF*. In *GhHSFB2a*-silenced cotton, one of the genes with encoding transcription factor bZIP was the most down-regulated DEG, with five times down-regulated expression. The most obvious up-regulated expression DEGs was one of the *AP2/ERF*, whose expression was up-regulated by up to 6.6-fold compared with pTRV2::*00* (Figure 6). Typically, these transcription factors are involved in disease resistance via plant hormones and secondary metabolites. For example, bZIP is closely related to ABA, and AP2/ERF is closely related to ethylene. These results suggest that *GhHSFB2a* regulates plant hormones and secondary metabolites in response to disease resistance.

### 2.6. Gene Ontology (GO), Kyoto Encyclopedia of Genes and Genomes (KEGG) and Protein-Protein Interaction (PPI) Analyses Revealed GhHSFB2a Functions

In order to further explore the mechanism of *GhHSFB2a* in cotton disease resistance, the data of transcriptome were enriched through GO, KEGG, and PPI to analyze the pathways involved in *GhHSFB2a*. The GO analysis revealed that DEGs were enriched in a variety of ABA-related metabolic pathways (“oxidoreductase activity, acting on single donors with incorporation of molecular oxygen, incorporation of two atoms of oxygen” “dioxygenase activity” “oxidoreductase activity, acting on single donors with incorporation of molecular oxygen”) (Figure 7a). KEGG analysis showed that differential genes were enriched in pathways closely related to the synthesis of secondary metabolites (“Phenylpropanoid biosynthesis”, “Linoleic acid metabolism”) and ABA synthesis (“Carotenoid biosynthesis”) (Figure 7b). PPI analysis showed that there were 47 interactions between proteins of pathways related to ABA metabolism, and six major node proteins were found in the PPI network by degree, namely LOC2-ODα1 (LOC107951462), LOCSAPK3like (LOC107886411), LOCSAPK2 (LOC107913412), LOCα-DOX2 (LOC107899049), LOCIPPase (LOC107907408), and LOCIMPase3 (LOC107925171). In addition to participating in pathways related to ABA metabolism, these nodal proteins are also involved in “peroxidase activity,” “antioxidant activity,” “response to oxidative stress” pathways which related to disease resistance (Figure 7c). Based on these observations, *GhHSFB2a* could play a significant role in disease resistance by regulating the ABA metabolic pathway.

### 2.7. Gene Expression Related to Plant Hormone and Secondary Metabolites in Cotton Was Induced by V. dahliae

To determine whether the expression of marker genes in the pathway related to plant hormones and secondary metabolites was induced by *V. dahliae*, cotton was infected with V991 and sampled at 0, 2, 12, and 72 hpi. The expression levels of marker genes selected from ABA (*GhNCED3*, LOC107918579; *GhNCED6*, LOC107890583), ethylene (*GhERF114*, LOC 107925292; *GhERF061*, LOC107947116), linoleic acid (*GhLox1*, LOC107919998), and phenylpropane (*GhCAD1*, LOC107953373) were detected using a qRT-PCR. It was found that these marker genes were expressed to different degrees. *GhERF114* was induced to the greatest extent, and its gene expression was about six times as much as that of the control (Figure 8). These results indicate that ABA, ethylene, linoleic acid, and phenylpropane-related marker genes in cotton are induced in response to the infection caused by *V. dahliae*.

### 2.8. Overexpression of GhHSFB2a Enhanced V. dahliae Resistance in Arabidopsis

To further confirm the role of *GhHSFB2a* in resistance to *V. dahliae*, *GhHSFB2a* was overexpressed (OE) in *A. thaliana* using the flower-dipping method. The results show that the target band could not be amplified from the DNA of WT *A. thaliana* and the negative control, but approximately 950 bp amplification products were detected in the DNA of overexpressed transgenic lines (Figure 9a). Among the five positive lines, OE-2 and OE-3, two lines with higher expression levels, were screened using a qRT-PCR and were selected for subsequent experiment. (Figure 9b). After inoculation of the WT and transgenic lines with V991, the WT leaves had more severe etiolation and wilting phenotypes than OE-2 and OE-3, and fungal biomass was significantly reduced (Figure 9c,d). These results indicate that ectopic expression of *GhHSFB2a* enhanced resistance to *V. dahliae* in *A. thaliana*.

## 3. Discussion

Plants respond to abiotic and biotic stresses through developmental, physiological, and biochemical pathways that are regulated by a network of transcription factors, including HSFs. HSFs are important factors in the regulation of plant stress resistance and play a vital role in plant stress response, growth, and development regulation [7]. Studies have shown that HSFB is also involved in the regulation of plant immunity, but the mechanism is not clear. In our previous study, *A. thaliana* inoculated with *V. dahliae* was sequenced using transcriptomics and metabolomics. We found that one of the DEGs was significantly upregulated by *V. dahliae*. It is homologous to *G. hirsutum GhHSFB2a*; the *V. dahliae*-induced expression pattern of *GhHSFB2a* rose at 0.5 h and was significantly upregulated at 12 hpi. In addition, the promoter of *GhHSFB2a* contains a defense-reaction-associated cis-acting element-FC-rich motif. These results imply that *GhHSFB2a* may play a role in cotton plant response to *V. dahliae*.

The *G. hirsutum* homologous gene, *GhHSFB2a*, belongs to class B of the HSFs family. In terms of protein structure, there is no AHA motif at the C-terminus of *GhHSFB2a*, which means that *GhHSFB2a* cannot independently complete transcriptional activation. Studies have shown that tomato *HSFB1* has a synergistic effect with class A HSFs, and the two combine to form a complex similar to an enhancer, which recruits plant histone HAC1 to form a ternary complex, thus synergistically activating reporter gene expression [10]. We hypothesized that *GhHSFB2a* may form a complex with other transcription factors and exert its transcriptional function to participate in cotton disease resistance.

HSFs are involved in the regulation of plant immunity [26,27]. Pick et al. demonstrated that *AtHsfB1a* also plays a pivotal role in primed defense gene activation and pathogen-induced acquired immune responses [28]. *CaHsfB2a* positively regulates plant immunity against *Ralstonia solanacearum* inoculation and tolerance to high temperature and humidity via transcriptional cascades and positive feedback loops involving *CaWRKY6* and *CaWRKY40* [29]. Phylogenetic tree analysis indicated that it might have functions similar to those of HSFB2a in *F. vesca*. The FvHSFB2a protein is located in the nucleus and has been demonstrated to be related to pathogen resistance [30,31]. Likewise, *GhHSFB2a* is also located in the nucleus. 

VIGS refers to the process in which a virus carrying a target gene fragment infects a plant and induces endogenous gene silencing and is currently widely used in plants for the analysis of gene function [32]. In *Manihot esculenta*, VIGS was used to treat autophagy-associated genes *MeATG8s* (*MeATG8a, MeATG8f, MeATG8h*), and the expression levels of these genes were significantly decreased, which results in the reduced resistance of plants to *Xanthomonas axonopodis* pv. *manihotis* [33]. In *Cucumis sativus* L., VIGS technology was used to instantaneously silence chitinase (chi) genes *chi2* and *chi14*, which impaired the tolerance of cucumber to *Fusarium oxysporum* f. sp. *cucumerinum* and significantly downregulated resistance genes related to the JA pathway [34]. To verify the function of *GhHSFB2a* in cotton resistance to *V. dahliae,* the VIGS analysis showed that knocking down the expression of *GhHSFB2a* and pTRV2::*GhHSFB2a* plants resulted in much more severe symptoms after *V. dahliae* inoculation compared to control plants. Overexpression is also an important method to confirm gene function, and overexpression of *GhHSFB2a* in *Arabidopsis* enhanced the resistance against *V. dahliae*, compared with the WT. These data imply that *GhHSFB2a* is a positive regulator and could be a key factor in cotton resistance to *V. dahliae.*

With the development of sequencing technology, gene function analysis using transcriptome sequencing has become a powerful research tool. In rice, transcriptome analysis has revealed that *OsPDCD5* regulates the biosynthesis of auxin, GA, and cytokinin and their related signaling pathways, thus affecting plant growth, reproduction, and response to external stimuli [35]. In this study, a transcriptome analysis revealed that the genes related to plant disease resistance, including *WRKY*, *MYB*, *bZIP*, and *AP2/ERF*, were significantly differentially expressed after *GhHSFB2a* silencing compared to the control. At present, a large amount of evidence shows that these transcription factors are involved in disease resistance through plant hormones and secondary metabolites. For example, the *CaWRKY27* gene from *Capsicum annuum* was overexpressed in *N. benthamiana*, and *CaWRKY27* positively regulated the resistance response of *N. benthamiana* to *R. alstonia solanacearum* infection through an ethylene-mediated signaling pathway [36]. MYB is a key regulatory factor for the synthesis of phenylpropane-derived compounds in plants, and phenylpropyl derivatives play an important role in plant defenses against biological stress [37]. Further analysis using GO, KEGG, and PPI showed that DEGs were significantly enriched in pathways related to ABA, ethylene, linoleic acid, and phenylpropanoid. In rice, *ONAC066* is involved in resistance to rice blast and bacterial leaf blight by regulating ABA signaling pathways [38]. *Sly*miR482e-3p mediates tomato wilt disease by modulating the ethylene response pathway [39]. Linoleic acid can induce systemic resistance in *N. benthamiana* to *Pectobacterium carotovorum* subsp. *carotovorum* [40]. In summary, we speculate that *GhHSFB2a* participates in cotton disease resistance by regulating ABA, ethylene, linoleic acid, and phenylpropanoids.

## 4. Materials and Methods

### 4.1. Bioinformatic Analysis

The HSFB protein sequences of *G. hirsutum* and other species were retrieved from the National Center for Biotechnology Information (https://blast.ncbi.nlm.nih.gov/) website (accessed on 6 July 2022), and MEGA7.0 was used for phylogenetic analysis. Multiple sequence alignments were performed using ESPript (https://espript.ibcp.fr/ESPript/ESPript/index.php) (accessed on 15 July 2022). The tertiary structure of the protein was analyzed using AlphaFold (https://alphafold.ebi.ac.uk/) (accessed on 18 July 2022). We used a 2000 bp region located upstream of *GhHSFB2a* to analyze cis-acting elements using the PlantCARE database (https://bioinformatics.psb.ugent.be/webtools/plantcare/html/) (accessed on 22 July 2022). TBtools was used for making heat map and visualize relative expressions.

### 4.2. Growth of Plant Material and Pathogen Cultures

Upland cotton (*G. hirsutum*) seeds were planted in a mixture of vermiculite and nutrient soil (1:2, *w*/*w*) and cultured at a constant temperature of 25 °C under long-day conditions with an 8/16 h dark/light photoperiod and a relative humidity of 60%. *Arabidopsis* seedlings were transplanted to nutrient-supplemented soil and cultured under an 8/16 h dark/light cycle at 22 °C and 60% relative humidity.

The highly toxic and defoliating pathogenic *V. dahliae* strain (V991) was cultured on potato dextrose agar (PDA, potato 200 g/L, glucose 20 g/L, agar 20 g/L) containing kanamycin 100 μg/mL and ampicillin 100 μg/mL for seven days at 25 °C. The fungus was then inoculated into complete medium (CM, yeast extract 6 g/L, casein acid hydrolysate 6 g/L, sucrose 10 g/L) at 25 °C and 220 rpm for another three d until the concentration of spores reached approximately 10^7^ CFU/mL [41].

### 4.3. Gene Expression Analysis

To determine the expression profile of *GhHSFB2a* involved in the plant defense response, cotton roots were dipped in a suspension of V991 spores for 5 min, and the total RNA from roots, stems, and leaves was extracted at 0, 0.5, 1, 2, 4, 6, 8, 12, 24, 48, and 72 hpi. First-strand cDNA was generated from 1 μg of total RNA using the HiScript III 1st Strand cDNA Synthesis kit (Vazyme, Nanjing, China). qRT-PCR primers for *GhHSFB2a* were designed (Appendix A), and *Gossypium hirsutum polyubiquitin* (*GhUbq*) served as the housekeeping gene (Appendix A). qRT-PCR was performed in 10 μL reactions using ChamQ SYBR qPCR Master Mix (Vazyme, Nanjing, China) and ABI 7500 Fast Real-Time PCR (Applied Biosystems, Foster City, CA, USA). The expression level of each target gene was analyzed using the 2^−ΔΔCT^ method [42].

### 4.4. Subcellular Localization and Confocal Microscopy

WoLF PSORT analysis (https://wolfpsort.hgc.jp/) (accessed on 2 August 2022) indicated that *GhHSFB2a* is localized to the nucleus. To determine the subcellular localization of *GhHSFB2a*, the full-length *GhHSFB2a* cDNA was amplified and introduced into the pYBA1132 vector to generate the pYBA1132::*GhHSFB2a*::*eGFP* construct. *A. tumefaciens* strain EHA105 containing pYBA1132::*GhHSFB2a*::*eGFP* or pYBA1132::*eGFP* (control) and pYBA1132::*H2B*::*mCherry* was co-injected into *N.benthamiana* leaves, and the signals were observed under a fluorescence confocal microscope (ZEISS LSM980 with Airyscan2) at 48 h post-infiltration.

### 4.5. VIGS and RNA-seq Analysis

The part of coding sequence (CDS, 400 bp) of *GhHSFB2a* was amplified via PCR using the Coker 312 cotton cDNA and pTRV2-GhHSFB2a-F/R primers (Appendix A). The fragment was inserted into the pTRV2 vector (provided by Professor Yule Liu, Tsinghua University, China), and infusion technology was used to construct the pTRV2::*GhHSFB2a* recombinant plasmid, which was used to transform *A. tumefaciens* GV3101 strain. For the VIGS assay, equal volumes of *A. tumefaciens* harboring pTRV1 and pTRV2::*GhHSFB2a* or pTRV2::*00* (control) were mixed, the cotyledons of seven-day-old cotton seedlings were infiltrated with a needleless syringe, and pTRV2::*CLA1* was used as the positive control.

After VIGS treatment, when the newly emerged leaves exhibited a photobleaching phenotype, the total RNA from the cotton leaves was extracted, and the silencing efficiency was detected using qRT-PCR. VIGS-silenced plants were sequenced using RNA-seq, and pTRV2::*00* was used as the control. Cultures and cells were handled as previously described [42]. RNA was extracted using the RNAprep Pure Plant Plus Kit (Polysaccharides & Polyphenolics-rich) (QIAGEN, Hilden, Germany), and the quality of the RNA was determined using an Agilent 2100 BioAnalyzer (Agilent Technologies, Santa Clara, CA, USA). Transcriptome sequencing libraries were constructed using the NEBNext^®^ Ultra™ RNA Library Prep Kit for Illumina^®^. Qubit 2.0 Fluorometer and qRT-PCR were used to detect the quality of the library. The Illumina NovaSeq 6000 (Illumina, San Diego, CA, USA) platform was used to generate 150 bp long paired-end reads. Fastp (version 0.19.7) was used to filter Rawdata and obtain Cleandata. HISAT2 V2.0.5 software was used to compare paired terminal clean reads with *G. hirsutum* TM-1, and the gene expression was quantitatively analyzed using Subread. Using DESeq, we obtained DEGs that were screened using |log_2_ (Fold Change)| > 1 and padj < 0.05. ClusterProfiler was used to perform GO and KEGG enrichment analysis of DEGs. Cytoscape was used for PPI analysis and visual analysis. 

### 4.6. A. thaliana Transformation and Molecular Analysis

The CDS of *GhHSFB2a* was amplified from Coker 312 cotton cDNA using overexpression primers (Appendix A). To construct a plasmid for overexpression, the fragment of *GhHsfB2a* was cloned into the entry vector pCAMBIA2300-35S-e green fluorescent protein (GFP) using an infusion reaction. *A. tumefaciens* strain GV3101, containing the *GhHSFB2a* overexpression plasmid, was used to transform *Arabidopsis* Col-0. *A. tumefaciens* with the overexpression plasmid was cultured to OD_600_ = 0.8, resuspended in 0.05% Silwet I-77 and 5% sucrose. *A. thaliana* was incubated for five weeks, and the flowers were immersed in the resuspended solution to obtain overexpressed plants. T_0_ generation seeds were cultured in MS screening medium containing NPTII resistance, and T_3_ transgenic lines were obtained after multiple screenings. Genomic DNA from WT and overexpressing plants was extracted to screen for positive genes using PCR amplification with detection primers (Appendix A). qRT-PCR of *GhHSFB2a* gene expression levels in overexpressing plants was performed as described previously.

### 4.7. Pathogenicity Assay and Determination of Fungal Biomass

To determine the performance of cotton in response to *V. dahliae*, three-leaf-stage seedlings were inoculated with V991. The plant disease index (DI) was calculated as previously described [43]. *A. thaliana* infection with *V. dahliae* was performed according to the style of previous methods. [44]. Plant fungal biomass was calculated as previously described [43].

To detect fungal biomass, plant roots were collected 14 dpi for genomic DNA extraction. Internal transcribed spacer (ITS) was used to quantify fungal colonization in the fragment amplified from *V. dahliae* using ITS-F/R (Appendix A), and the cotton *GhUbq* gene served as an endogenous plant control. Fungal biomass was determined using qRT-PCR, as described previously [41].

### 4.8. Statistical Analysis

All experiments were performed in triplicate. Statistical data were analyzed using the IBM SPSS Statistics for Windows (Version 20.0). One-way analysis of variance (ANOVA) with Duncan’s multiple range test were used to analyze the data. Differences were considered statistically significant at *p* < 0.05.

## 5. Conclusions

In this study, the *GhHSFB2a* gene played a critical regulatory role in plant disease resistance pathways. The expression of *GhHSFB2a* was considerably upregulated at 0.5 hpi and peaked at 12 hpi. The GhHSFB2a protein was localized in the nucleus. The VIGS analysis showed that silencing *GhHSFB2a* increased fungal colonization and reduced resistance against *V. dahliae*. The transcriptome and maker genes analysis showed that *GhHSFB2a* regulated cotton disease resistance through ABA, ethylene, linoleic acid, and phenylpropanoids (Figure 10). Overexpression of *GhHSFB2a* enhanced *V. dahliae* resistance in *Arabidopsis.* In summary, the results indicate that *GhHSFB2a* is involved in the resistance of cotton to *V. dahliae*, which provides new strategies for the creation of new highly resistant cotton materials.

## Figures and Tables

**Figure 1 ijms-24-01845-f001:**
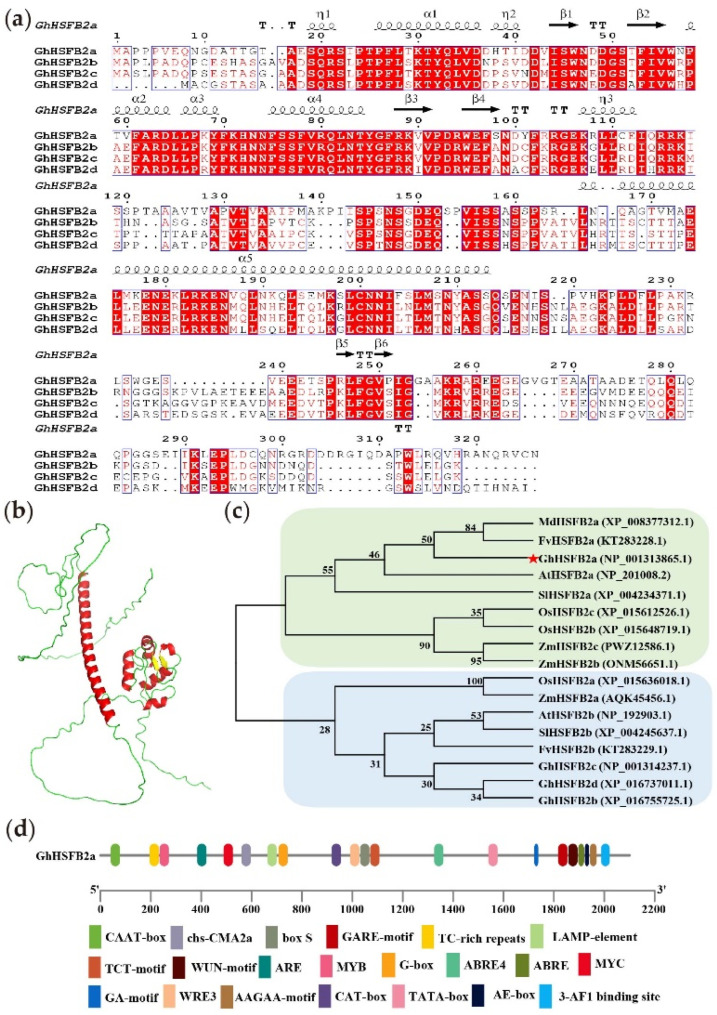
Structural characteristics of *GhHSFB2a*. (**a**) Multiple sequence alignment of GhHSFB2a, GhHSFB2b, GhHSFB2c, and GhHSFB2d proteins. The common residues of the four sequences are displayed in white letters on a red square, and similar residues are shown in red. The black dot represents a single-amino-acid deletion, and the black arrow and coiled helix represent the predicted secondary structure. (**b**) Three-dimensional GhHSFB2a model. (**c**) Phylogenetic tree analysis of HSFB2 proteins from *Gossypium hirsutum* (Gh), *Solanum lycopersicum* (SI), *Arabidopsis thaliana* (At), *Fragaria vesca* (Fv), *Malus domestica* (Md), *Oryza sativa* (Os), *Zea mays* (Zm). (**d**) Cis-acting element analysis of *GhHSFB2a* promoter region.

**Figure 2 ijms-24-01845-f002:**
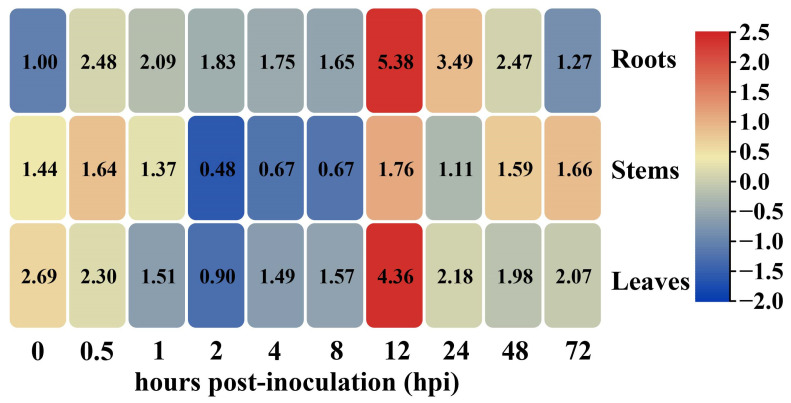
Expression of *GhHSFB2a* in cotton inoculated with *V. dahliae.* Heat map analysis of *GhHSFB2a* gene expressions in different organs (roots, stems, and leaves) and different inoculation time (0, 0.5, 1, 2, 4, 8, 12, 24, 48, and 72 hpi) with *V. dahliae* of Coker 312 cotton. The color from blue to red indicates low to high expression.

**Figure 3 ijms-24-01845-f003:**
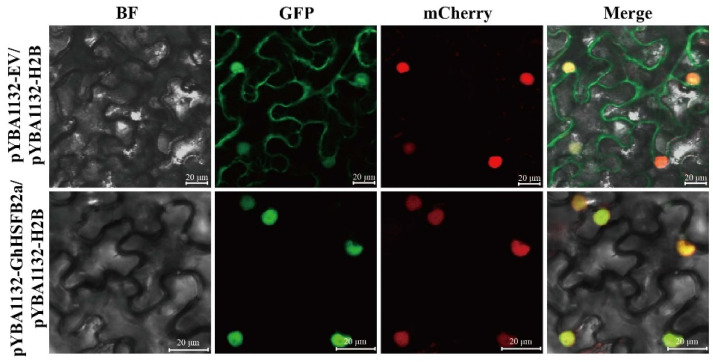
Subcellular localization of GhHSFB2a. *Agrobacteria* carrying the pYBA1132-*green fluorescent protein* (*GFP*) or pYBA1132*-GhHSFB2a-GFP* vector were inserted into the leaves of four-week-old *N. benthamiana* plants expressing the nuclear-localized marker H2B-mCherry. BF, the bright field. GFP protein was localized both in cytoplasm, plasma membrane and in the nucleus, and GFP-fused GhHSFB2a protein was localized in the nucleus. Results were visualized using a confocal microscopy 48 h after transformation. Bars = 20 μm.

**Figure 4 ijms-24-01845-f004:**
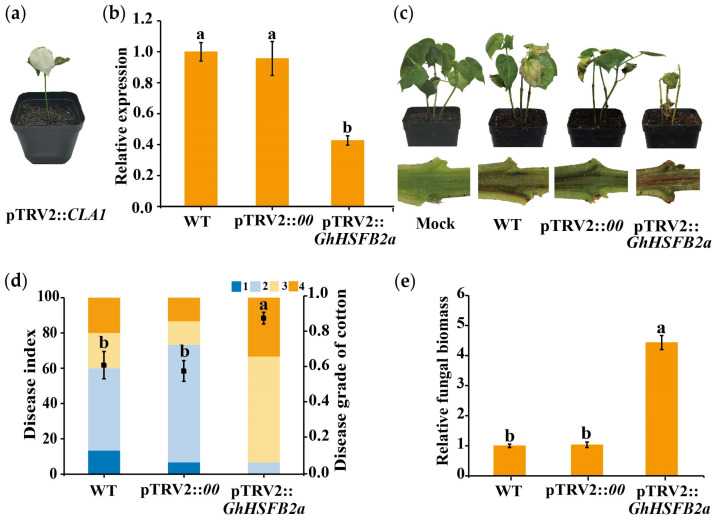
Silencing of *GhHSFB2a* attenuates plant resistance to *V. dahliae*. (**a**) Seven-day-old cotton plants were infiltrated with *Agrobacterium tumefaciens* carrying pTRV2::*CLA1*. The photographs were taken two weeks after infiltration. (**b**) Preliminary assay of the efficiency of VIGS under our experimental conditions. Expression of *GhHSFB2a* in pTRV2::*GhHSFB2a* and control plants. *Gossypium hirsutum polyubiquitin* was the internal reference gene (*p* < 0.05; 2^−ΔΔCt^). (**c**) Disease phenotypes of *GhHSFB2a*-silenced plants infected with V991 at 14 days post-inoculation (dpi). (**d**) Disease index of pTRV2::*GhHSFB2a* and control plants at 14 dpi with *V. dahliae*. 1, 2, 3, and 4 represent the disease degree. (**e**) Fungal biomass determined using qRT-PCR at 14 dpi with *V. dahliae*. Error bars represent the standard deviations and different letters indicate significant differences at *p* < 0.05.

**Figure 5 ijms-24-01845-f005:**
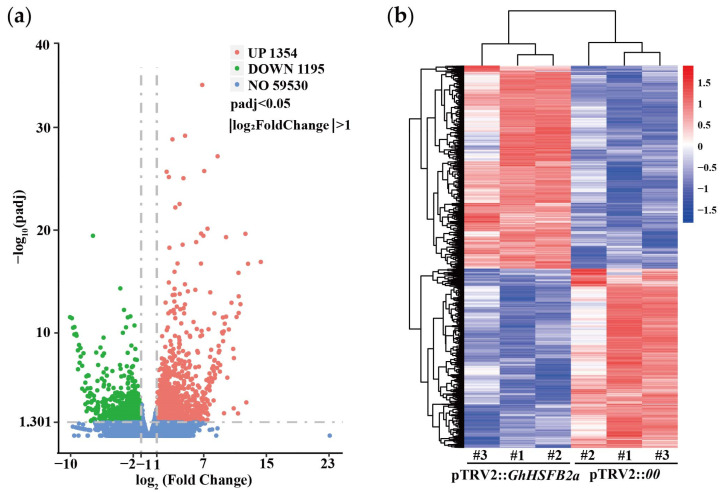
Analysis of DEGs in pTRV2::*00* and pTRV2::*GhHSFB2a*. (**a**) Volcano plot, identification map of DEGs using Log_2_ (Fold Change) > 1 and padj < 0.05. (**b**) Heatmap of DEGs of each sample.

**Figure 6 ijms-24-01845-f006:**
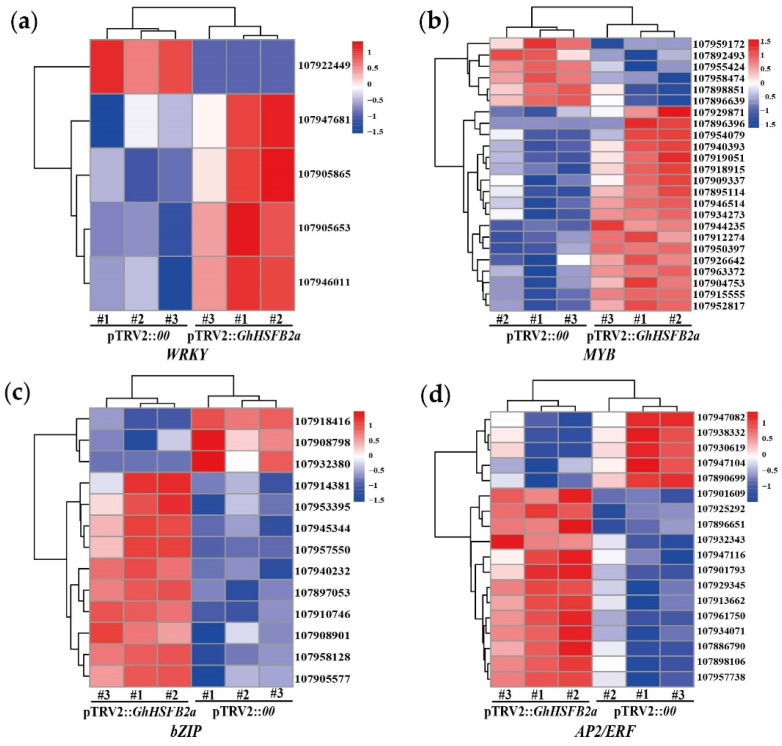
Expression of transcription factor genes associated with plant resistance in pTRV2::*00* vs. pTRV2::*GhHSFB2a*. Heat maps indicating expression comparison of enzyme genes or transcription-factors among four comparison sets. (**a**) WRKY. (**b**) MYB. (**c**) bZIP. (**d**) AP2/ERF. Red represents up-regulation and blue represents down-regulation.

**Figure 7 ijms-24-01845-f007:**
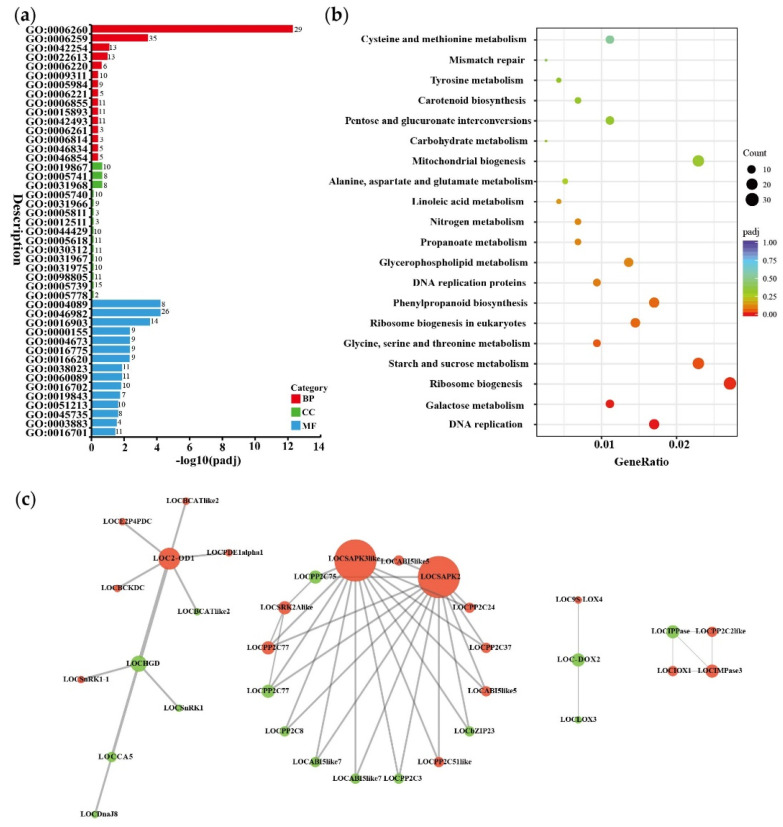
GO, KEGG enrichment, and PPI analysis for DEGs in pTRV2::*00* and pTRV2::*GhHSFB2a* cotton. (**a**) GO classification of DEGs in pTRV2::*00* vs. pTRV2::*GhHSFB2.* BP, biological process; CC, cell component; MF, molecular function. (**b**) KEGG enriched bubble map of DEGs in pTRV2::*00* vs. pTRV2::*GhHSFB2a*. (**c**) PPI analysis of DEGs in pTRV2::*00* vs. pTRV2::*GhHSFB2a*. The size of the nodes was set according to their degree using Cytoscape. Red nodes represent upregulated genes and green nodes represent downregulated genes.

**Figure 8 ijms-24-01845-f008:**
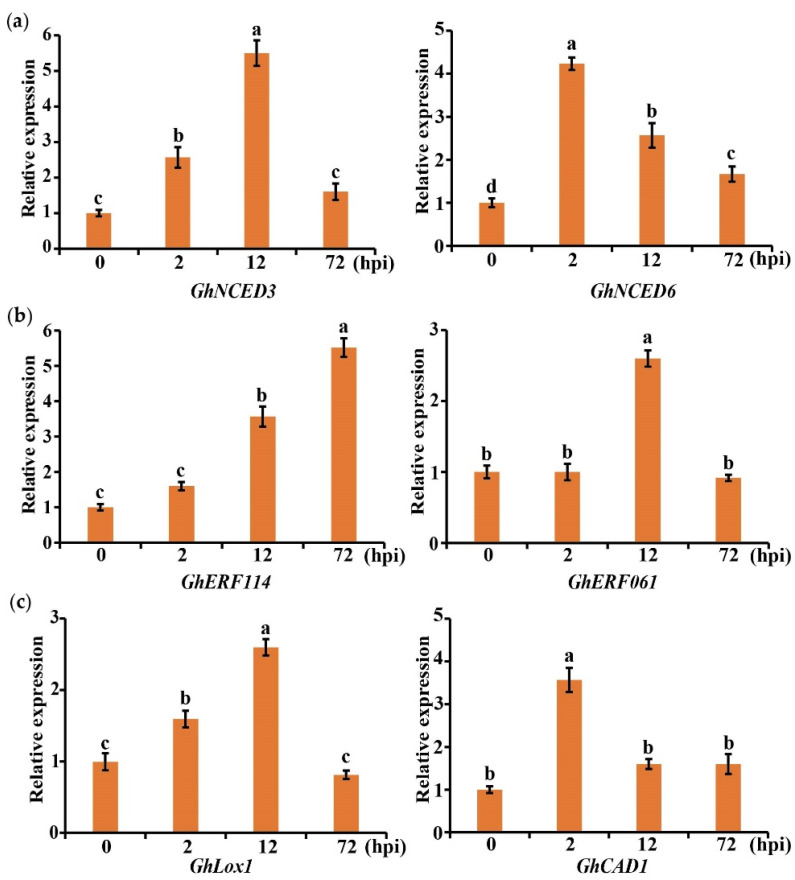
Marker genes were used to detect whether ABA, ethylene, and secondary metabolite-related pathways were involved in the response of cotton to *V. dahliae*. (**a**) Marker genes in ABA-related pathways. (**b**) Marker genes in ethylene-related pathways. (**c**) Marker genes in secondary metabolite-related pathways. Error bars represent the standard deviations and different letters indicate significant differences at *p* < 0.05.

**Figure 9 ijms-24-01845-f009:**
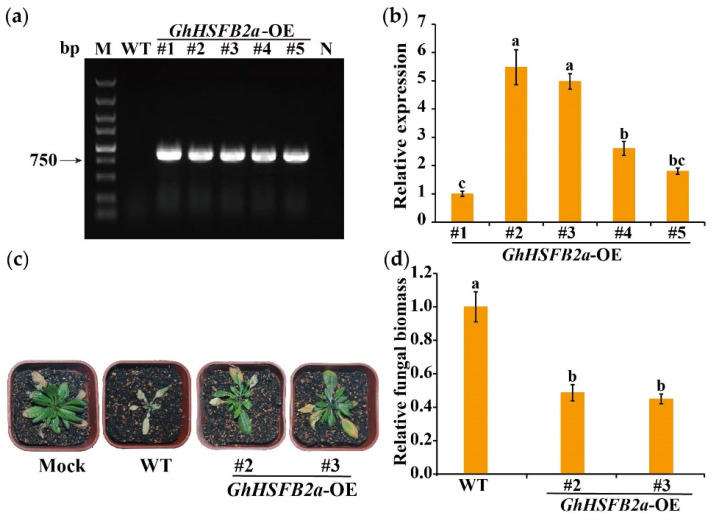
Enhanced *V. dahliae* resistance in *A. thaliana* plants overexpressing *GhHSFB2a*. (**a**) Positive transformants were identified using PCR amplification. M, maker; N, negative control. (**b**) Expression levels of *GhHSFB2a* driven by the 35S promoter in transgenic *A. thaliana* lines (L1, L2, L3, L4, and L5). Actin was used as an internal control. (**c**) Symptoms in WT and *GhHSFB2a* transgenic *Arabidopsis* plants inoculated with *V. dahliae*. (**d**) Fungal biomass determined using a qRT-PCR in WT and transgenic *Arabidopsis* plants. Error bars represent the standard deviations and different letters indicate significant differences at *p* < 0.05.

**Figure 10 ijms-24-01845-f010:**
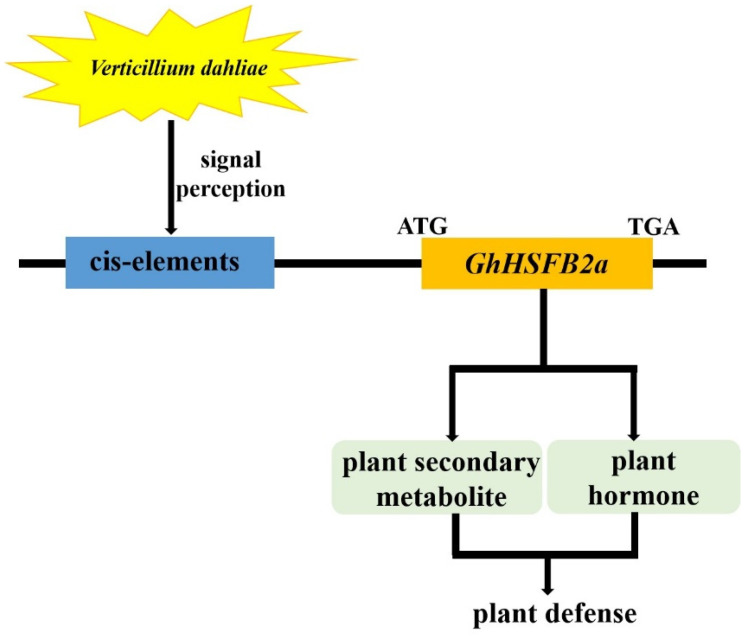
Model for *GhHSFB2a* function in cotton. *GhHSFB2a* regulated plant defense by affecting plant phytohormones (ethylene, ABA) and secondary metabolic (linoleic acid, phenylpropanoids).

## Data Availability

The original contributions presented in the study are publicly available. This data can be found here: NCBI, PRJNA911590.

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
