# Peer review of "Heat Shock Transcription Factor GhHSFB2a Is Crucial for Cotton Resistance to Verticillium dahliae"

_ijms, 2023, doi:10.3390/ijms24031845_

Round 1

Author Response

Dear Reviewer,

It is in response to your suggestions regarding our manuscript entitled “Heat shock transcription factor GhHSFB2a is crucial for cotton resistance to Verticillium dahliae” (ijms-2136137). We would like to thank you on behalf of all the authors for reviewing our manuscript. We have responded to all the suggested revisions point-by-point as below.

Reviewer 2 Report

Cotton (Gossypium hirsutum) is the most important natural textile fiber worldwide, accounting for approximately 35% of the annual global fiber demand. The losses in cotton yield and quality are mainly caused by Verticillium wilt (VW), a vascular disease induced by the soil-borne pathogen Verticillium dahliae. In this study, the authors identified and functionally characterized GhHSFB2a, which is a DEG belonging to HSFs and related to cotton immunity to V. dahliae. As a whole, the data is solid and the conclusions are proper. However, some details should be revised. Below some minor suggestions are listed:

1.     The full name shall be marked when the Latin name appears for the first time, and the abbreviation shall be marked when the Latin name appears for the second time or more. For example, in line 58, should be changed Arabidopsis to Arabidopsis thaliana. Line 387, please change Arabidopsis thaliana to A. thaliana. Line 126, please change N. benthamiana as Nicotiana benthamiana. Line 360, please change Nicotiana benthamiana to N. benthamiana.

2.     Please add the space between the symbol and the number. Line 113, please add a space between the dash and the number of "0–72hpi".

3.     Line 113, please change hpi to hours post indentation (hpi). Line 347, please change hours post indentation (hpi) to hpi.

4.     Line 124. “GhHSFB2a” should not be italic.

5.     Hyphens do not need to be italicized. Line 136, change the hyphen of “pYBA1132-green fluorescent protein” to regular font, and please remove the space between “-” and “green”.

6.     Line 140: “pYBA1132-GhHSFB2a-GFP” is a constructed plasmid and cannot be expressed as a functional protein. And “at” should be changed to “in”.

7.     Line 161, please change A. tumefaciens to Agrobacterium tumefaciens. Line 358, please change Agrobacterium tumefaciens to A. tumefaciens.

8.     Line 163, GhUbq should be modified to Gossypium hirsutum polyubiquitin.

9.     Line 257-262: The corresponding references need to be added.

10.  Line 283. “FvHSFB2a should not be italic.

11.  Line 310-312: Expression needs to be adjusted. Do you want to express genes or pathways?

12.  Line 341: “d” should be changed to “days”.

13.  Line 364: “The CDS (400 bp)” does not represent the full length of the CDS, but a partial sequence.

14.  Line 405. “Fungal Biomass” should be lowercase.

Author Response

Dear Reviewer,

Thank you on behalf of all the authors for your review of “Heat shock transcription factor GhHSFB2a is crucial for cotton resistance to Verticillium dahliae” (IJMS-2136137). We responded to all the suggested revisions point-by-point as below.

Reviewer 3 Report

Heat shock transcription factors (HSFs) play a critical regulatory role in many plant disease resistance pathways. However, the molecular mechanisms of cotton HSFs involved in resistance to the soil-borne fungus Verticillium dahliae are limited. The study was conducted using bioinformatics, gene expression analysis, cotton VIGS, overexpression of Arabidopsis thaliana and RNA-seq techniques demonstrated that GhHSF2a was resistant to V. dahliae and its mechanism. The study has been well designed and conducted. I would suggest some minor change before publication.

1.      PPI are also included in the results, so it is recommended to add PPI analysis in the materials and methods and discussion parts

2.      Line 18. Would it be nice to change “HSFB2” to “type 2 of HSFB”.

3.      Line 23. Please add the abbreviation for “hours post-inoculation” and line 24 the abbreviation for “Virus-induced gene silencing”.

4.        Line 28,73,88: “GhHSFB2a” should not be italic.

5.        Line 29-30: “Overexpression of Arabidopsis GhHSFB2a gene showed a significant increase in disease resistance” There is a problem with the expression of this sentence, please readjust it.

6.        Line 57: Tenses are inconsistent with previous expressions and need to be adjusted.

7.      Line 67, 75 and 81. “resistance to V. dahliae” come up too often.

8.      Line 73. Please correct “GhHSFB2a” to “GhHSFB2a”.

9.      Line 86. “ which contains……” and Line 88 “ GhHSFB2a contains……” sentence pattern repetition.

10.    Line 105-106:GhHSFB2a, GhHSFB2b, GhHSFB2c, and GhHSFB2d should not be italic.

11.  Line 122. Please describe the -lg (FPKM) of the heat map.

12.  Line 130. Please correct “nucleus,which” to “nucleus, which”.

13.  No space is required between hyphens. Line 166, please remove the space in the “qRT-PCR”.

14.  Line 178. “pTRV2::GhHSFB2a-silenced” semantic repetition.

15.  Line 194, please change GO and KEGG to Gene Ontology (GO) and Kyoto Encyclopedia of Genes and Genomes (KEGG). For lines 385-386, please change Gene Ontology (GO) and Kyoto Encyclopedia of Genes and Genomes (KEGG) to GO and KEGG respectively.

16.  Line 325. Please supplement heat map making software.

17.  Please complete the name of the fusion protein. Line 359, it is better to change pYBA1132::H2B to pYBA1132::H2B::mCherry.

Author Response

Dear Reviewer,

Thank you on behalf of all the authors for your review of “Heat shock transcription factor GhHSFB2a is crucial for cotton resistance to Verticillium dahliae”. We responded to all the suggested revisions point-by-point as below.
